# Complementary Targeting of Rb Phosphorylation and Growth in Cervical Cancer Cell Cultures and a Xenograft Mouse Model by SHetA2 and Palbociclib

**DOI:** 10.3390/cancers12051269

**Published:** 2020-05-17

**Authors:** Amy L. Kennedy, Rajani Rai, Zitha Redempta Isingizwe, Yan Daniel Zhao, Stanley A. Lightfoot, Doris M. Benbrook

**Affiliations:** 1Department of Pathology, College of Medicine, University of Oklahoma Health Sciences Center, Oklahoma City, OK 73104, USA; Amy-Bosley@ouhsc.edu; 2Stephenson Cancer Center, University of Oklahoma Health Sciences Center, Oklahoma City, OK 73104, USA; Rajani-Rai@ouhsc.edu; 3Department of Pharmaceutical Sciences, College of Pharmacy University of Oklahoma Health Sciences Center, Oklahoma City, OK 73104, USA; Zitha-Isingiswe@ouhsc.edu; 4Department of Biostatistics and Epidemiology, College of Public Health University of Oklahoma Health Sciences Center, Oklahoma City, OK 73104, USA; daniel-zhao@ouhsc.edu; 5Center for Cancer Prevention and Drug Development, University of Oklahoma Health Sciences Center, Oklahoma City, OK 73104, USA; SL20008bl@outlook.com

**Keywords:** cyclin D1, retinoblastoma, palbociblib, SHetA2, cervical cancer

## Abstract

Cervical cancer is caused by high-risk human papillomavirus (HPV) types and treated with conventional chemotherapy with surgery and/or radiation. HPV E6 and E7 proteins increase phosphorylation of retinoblastoma (Rb) by cyclin D1/cyclin dependent kinase (CDK)4/6 complexes. We hypothesized that cyclin D1 degradation by the SHetA2 drug in combination with palbociclib inhibition of CDK4/6 activity synergistically reduces phosphorylated Rb (phospho-Rb) and inhibits cervical cancer growth. The effects of these drugs, alone, and in combination, were evaluated in SiHa and CaSki HPV-positive and C33A HPV-negative cervical cancer cell lines using cell culture, western blots and ELISA, and in a SiHa xenograft model. Endpoints were compared by isobolograms, ANOVA, and Chi-Square. In all cell lines, combination indexes documented synergistic interaction of SHetA2 and palbociclib in association SHetA2 reduction of cyclin D1 and phospho-Rb, palbociclib reduction of phospho-Rb, and enhanced phospho-Rb reduction upon drug combination. Both drugs significantly reduced phospho-Rb and growth of SiHa xenograft tumors as single agents and acted additively when combined, with no evidence of toxicity. Dilated CD31-negative blood vessels adjacent to, or within, areas of necrosis and apoptosis were observed in all drug-treated tumors. These results justify development of the SHetA2 and palbociclib combination for targeting phospho-Rb in cervical cancer treatment.

## 1. Introduction

While promising new molecularly targeted agents have been FDA-approved for other cancer types, there are none currently approved for first line cervical cancer therapy. Traditional chemotherapy, often combined with surgery and/or radiation, depending on the stage, is the contemporary standard of care for cervical cancer resulting in significant side effects [1,2]. In 2014, the US FDA approved addition of the anti-vascular endothelial growth factor (VEGF), angiogenesis inhibitor bevacizumab to chemotherapy for advanced cases of cervical cancer only [3]. A decrease in the cervical cancer death rate by more than half in 2016 compared to 1975 is attributed to implementation of screening programs that have reduced incidence and increased early detection [4]. However, between 2007 and 2016 the decrease in cervical cancer death rate leveled out to only 1% per year in women over the age of 50 with no decrease in women less than 50 years old [4,5]. The success of the screening programs is due to the detection and removal of dysplastic lesions caused by human papilloma virus (HPV) infection. There are over one hundred types of HPV which can be categorized into high or low risk for causing cervical cancer [6]. High risk HPV examples include HPV-16 and HPV-18, while low risk examples include HPV-6 and HPV-11 [5]. When allowed to persist, lesions harboring high risk HPV types can progress to cancer.

HPV carcinogenesis is driven by overexpression of the HPV Early 6 and 7 (E6 and E7) proteins, which increase cell proliferation and genomic instability by binding and inactivating cellular tumor suppressor proteins [6] as illustrated in the Graphical Abstract. The HPV E7 protein functions in cervical cancer by inactivating the retinoblastoma (Rb) and p21 proteins [6,7]. Under normal conditions, Rb forms a complex with, and represses, E2F transcription factors in the G1 phase of the cell cycle. Phosphorylation of Rb by cyclin D1/cyclin dependent kinase 4/6 (CDK4/6) complexes causes release of E2F to transcribe cell cycle progression genes, which is required for G1 phase progression into S phase, and without this phosphorylation activity cells arrest in G1 phase [8]. Under normal circumstances, the p21 protein keeps proliferation under control by binding cyclin D1/CDK4/6 complexes to inhibit Rb phosphorylation and E2F release, thereby inhibiting cell cycle progression [9]. The HPV E7 protein contains an LXCXE sequence motif, which mediates binding to Rb leading to Rb degradation, release of E2F from Rb repression and acceleration of cell cycle progression [6]. The HPV E6 protein targets the wild-type p53 tumor suppressor protein, an inducer of p21, for degradation [10,11]. 

In most cancers the *TP53* gene is mutated, which causes increases in proliferation and genomic instability. However, most cervical cancers do not contain *TP53* mutations, most likely because E6 expression can substitute for the mutation. Thus, the HPV *E6* and *E7* genes cause increased cellular proliferation by reducing p53, p21 and Rb control over the cell cycle (Graphical Abstract). In cervical cancer, these HPV-driven molecular events justify targeting the down-stream cyclin D1/CDK 4/6 complexes in development of much-needed, new molecularly targeted agents.

A promising new drug currently entering a peer-reviewed clinical trial for cervical cancer is sulfur heteroarotinoid A2 (SHetA2, NSC 726189), a small molecule flexible heteroarotinoid (Flex-Het) which targets cyclin D1 for degradation [12]. SHetA2 binds to three heat shock protein A (HSPA) proteins (HSPA5, 8 and 9) resulting in G1 cell cycle arrest and mitochondria-mediated apoptosis in cancer cells, while the effect on healthy cells is limited to G1 cell cycle arrest [12,13,14]. The mechanism of SHetA2 cyclin D1 degradation involves phosphorylation, ubiquitination and proteasomal degradation [12] predicted to be caused by SHetA2-induced release of cyclin D1 from the HSPA chaperone proteins [15]. Expression of a non-phosphorylatable cyclin D1 mutant inhibited SHetA2-induced G1 cell cycle arrest confirming the role of cyclin D1 degradation in the SHetA2 mechanism [12]. 

To complement SHetA2 degradation of cyclin D1, we hypothesized that combination treatment with a CDK 4/6 inhibitor drug could block the activity of any remaining cyclin D1/CDK 4/6 complexes in a synergistic manner (Graphical Abstract). Palbociclib is a CDK 4/6 inhibitor currently used in the treatment of hormone receptor positive-metastatic breast cancers and being evaluated in clinical trials of multiple other cancer types [16]. The combination of these two drugs is predicted to have reduced side effects in comparison to current standard of care treatment for cervical cancer. Preclinical studies conducted by the National Cancer Institute found the no observed adverse event level (NOAEL) of SHetA2 in dogs to be >25-fold higher than the effective dose in cancer prevention and treatment studies [17,18,19]. The most common adverse event associated with palbociclib treatment is hematologic toxicity, mostly mild to moderate neutropenia, which can be reversed with dose reduction [20]. 

Our objectives were to evaluate the drug interaction effects and mechanisms of SHetA2 and palbociclib in cervical cancer cell lines and in an animal model. We hypothesized that the convergence of the SHetA2 and palbociclib mechanisms will be observed at Rb phosphorylation. Here, we provide cell culture and animal model data in support of this hypothesis.

## 2. Results

### 2.1. SHetA2 and Palbociclib Act Synergistically in Cervical Cancer Cell Lines

The efficacy of SHetA2 alone and in combination with palbociclib in HPV positive (HPV+/SiHa and CaSki) and HPV negative (HPV−/C33A) cervical cancer cell lines were evaluated using a cytotoxicity assay (MTT) and the Chou-Talalay method to determine if the drugs are additive, synergistic or antagonistic [21]. This method calculates the combination index (CI) and drug reduction index (DRI) using the dose and fold effects of the single and combined drugs. The CI determines if a drug combination has a synergistic (CI < 1), additive (CI = 1) or antagonistic (CI > 1) effect [22]. The DRI determines the fold reduction of a drug that can be used to accomplish a given effect when used in combination with the other drug in comparison to the dose of the drug required to achieve that effect level when administered alone [22]. To identify drug doses that will allow observation of interacting effects, the half maximal inhibitory concentration (IC_50_) value of each drug after 72 h of treatment was determined for each cell line (Table 1 and Figure 1 and Appendix A).

Chou-Talalay CIs calculated using the fold effects of a two-fold dilution series of the one-to-one ratios of the IC_50_s indicated that SHetA2 and palbociclib are synergistic in all three cell lines (Table 1, Figure 1 for representative experiments and Appendix A for replicate experiments). The DRI’s calculated for each cell line indicate that when used in combination, lower doses of the individual drugs can achieve the same effect level as the single drug treatments (Table 1). Overall, this data demonstrates syngergistic interaction between SHetA2 and palbociclib in cervical cancer cell lines independent of HPV status.

### 2.2. SHetA2 and Palbociclib Regulation of Cyclin D1 and Phosphorylation of Rb

Our hypothesized model of SHetA2 and palbociclib synergy predicts that SHetA2, but not palbociclib, will reduce cyclin D1 in treated cells. We anticipated that palbociclib would not further enhance cyclin D1 degradation. This expected pattern of cyclin D1 reduction was observed in the C33A cell line evaluated by two different methods, western blot (Figure 2A) and ELISA (Figure 2B). In Caski, both SHetA2 and palbociclib reduced cyclin D1 (Figure 2A,B). The level of cyclin D1 in SiHa was too low to allow accurate measurement of SHetA2 reduction in western blot or ELISA analysis, however, a western blot analysis confirmed that a low level of cyclin D1 is present in SiHa cells and can be induced to higher levels by nicotine-derived nitrosamine ketone (NNK) (Appendix A).

The synergistic point of our hypothesized model converges on the down-stream effects of SHetA2 and palbociclib on phospho-Rb levels. Western blot analysis demonstrated that both drugs reduced phospho-Rb levels in all three cell lines (Figure 3). The reduction was the greatest in the C33A cell line to the extent that the combination treatment was not significantly greater than the individual treatments. In CaSki cells, the combination treatment was significantly greater than either single drug treatment, while in the SiHa cells, the combination was only significantly greater than the SHetA2 single drug treatment. 

### 2.3. Additive Interaction of SHetA2 and Palbociclib In Vivo

To validate the drug interaction in vivo, female mice harboring SiHa xenograft tumors were treated with placebo, 60 mg/kg SHetA2, 100 mg/kg palbociclib, or the drug combination by oral gavage every day for 24 days. All three treatment groups exhibited significant reduction in tumor growth compared to the control group (ANOVA: SHetA2 *p* = 0.043, palbociclib *p* = 0.033, drug combination *p* = 0.014) (Figure 4). 

Because there appeared to be interactions between time and treatments, analyses were carried out by individual measurements using a two by two factorial design to determine if the effects of the drugs were additive when used in combination. A generalized estimating equations (GEE) model was generated using SHetA2 (*p* = 0.092), palbociclib (*p* = 0.234), time (*p* = 0.003), SHetA2 by time interaction (*p* = 0.040), and palbociclib by time interaction (*p* = 0.051). These p values indicate interactions within the model and not significant differences between the drug treatment groups and the control group. The results showed a significant additive effect of SHetA2 (*p <* 0.001) and palbociclib (*p <* 0.001) at time point 10. This additive effect between SHetA2 (*p* = 0.001) and palbociclib (*p* = 0.051) was trending significant at time point 9. Additionally, SHetA2 induced significant tumor suppressing activity as early as at time point 7 (*p* = 0.045).

There were no significant differences in the average body weights or growth of the mice during the treatment period indicating that the single or combined drug treatments did not cause gross toxicity to the animals (Figure 4B; Repeated Measures ANOVA *p* = 0.1572; Linear regression, 95% confidence intervals of growth rates: Control: 0.202 to 0.561 g/day, SHetA2: 0.079 to 0.703 g/day, palbociclib: 0.437 to 0.817 g/day, SHetA2+palbociclib: 0.294 to 0.814 g per day). Plasma specimens collected from the mice at the end of the experiment were tested for blood urea nitrogen (BUN) and creatinine (CREA) and kidney enzyme levels and for alkaline phosphatase (ALKP) and aspartate aminotransferase (AST) liver enzyme levels as an indication of potential drug toxicity. Enzymes levels higher than the max range are indicative of kidney or liver toxicity. All tests were low, which is not an indication of liver and kidney toxicity, or were within range, except for a slight AST elevation (266 U/L) in one mouse and very high AST elevation (442 U/L) in another mouse in the SHetA2 group (Table 2). 

Consistent with lack of toxicity of either drug, there were no significant differences in the organ-body weight ratios among the treatment groups (Table 3).

To validate the mechanism of drug action in vivo, tumors collected from the mice at the end of the experiment were evaluated with histologic and immunohistochemical stains (Figure 5). H&E stained tumor sections revealed necrosis in in all treatment groups. Tumor angiogenesis was evaluated with the trichrome stain and anti-CD31 immunohistochemical analysis. In the untreated group, blue blood vessels revealed by the trichrome stain were distributed throughout most of the non-necrotic areas of the tumors and stained positively for CD31. All treatment groups exhibited areas of dilated blue blood vessels that were localized within or adjacent to necrotic areas of the tumors. Interestingly, despite their large size, these abnormal blood vessels did not stain positively for CD31. Positive CD31 staining in a normal appearing blood vessel and negative CD31 staining in an abnormal dilated blood vessel within the same combination drug-treated tumor section confirms the robustness of the CD31 staining results. No obvious changes in the collagen or cell densities were observed in the stroma of trichrome stained sections. The living and necrotic areas of the tumors observed in the hematoxylin and eosin (H&E) stained sections were consistent with negative and positive staining for cleaved caspase 3 (active caspase 3), respectively, indicating occurrence of apoptosis in the necrotic areas. The opposite pattern was observed for phospho-Rb and cyclin D1 in that the non-necrotic areas stained positively, while the necrotic areas stained negatively.

The immunohistochemistry stain scores show that SHetA2 caused elevation of cleaved caspase 3 and reduced phospho-Rb, and that the combination treatment further enhanced these effects (Figure 6). Palbociclib alone reduced phospho-Rb staining, but did not alter cleaved caspase 3 staining. All of the tumors in the control and palbociclib groups had a cleaved caspase 3 stain score of 3 (intensity × percent positive cells), whereas 25% of the tumors in the SHetA2 group and 50% in the combination drug treatment group had higher stain scores of 6, however the numbers of tumors were too low for determination of statistical significance (Figure 6A). The cyclin D1 staining was too low for comparison between the groups. All of the tumors in the control group had phospho-Rb scores of 12 (intensity × percent positive cells), whereas there was a significant stepwise increase in the number of tumors with <12 stain scores in the SHetA2, palbociclib, and combination drug treatment groups that appeared to be additive between the two drugs (Figure 6B).

## 3. Discussion

While SHetA2 and palbociclib have been studied in multiple cancer types, this is the first study to evaluate the mechanisms, synergy and in vivo activity of these drugs in cervical cancer. Our findings validate the in vivo complementarity of SHetA2 and palbociclib against phospho-Rb and growth of cervical cancer xenograft tumors. Cell culture studies demonstrated a synergistic interaction between SHetA2 and palbociclib in association with enhanced reduction of phospho-Rb. We tested this interaction based on our hypothesized model (Graphical Abstract) that SHetA2-induced degradation of cyclin D1 is complemented by palbociclib inhibition of CDK4/6 kinase activity to more fully inhibit the cyclin D1/CDK4/6 complex phosphorylation of Rb in comparison to the singular activities of the individual drugs. Western blots of cell culture experiments confirmed that the synergy is associated with reduced cyclin D1 in SHetA2-treated cultures, and with reduced phosphorylation of Rb in cultures treated with SHetA2 and/or palbociclib. Consistent with the model, combined treatment with the two drugs did not increase the level of cyclin D1 degradation, but did enhance the level of Rb phosphorylation. In the validation animal model, the interaction of SHetA2 with palbociclib on tumor growth was significantly additive. Tumors from the combined treatment group trended to have increased reduction of Rb phosphorylation in comparison to the phospho-Rb levels observed in the tumors from single drug treatment groups.

It is likely that additional mechanisms contribute the cytotoxicities of SHetA2 and palbociclib when administered alone and in combination. SHetA2 causes upregulation of intrinsic and extrinsic apoptosis pathway components, degradation of Bcl-2 and translocation of wild type and mutant p53 to the nucleus and mitochondria [14,18,23,24,25,26,27]. Palbociclib has been shown to bind multiple off-target kinases and affect down-stream signaling in vitro and in vivo [28,29]. We did not evaluate sufficient numbers of HPV+ versus HPV− cell lines to make conclusions about the role of HPV in the drug synergy mechanism. However, the similar responses of the HPV− C33A cell line to the HPV+ cell lines suggests that the evolution of the originating tumors for these cell lines resulted in similar defects that make them susceptible to SHetA2 and palbociclib. Thus, the SHetA2 and palbociclib combination may have efficacy in non-HPV cancers.

Both SHetA2 and palbociclib appeared to reduce tumor angiogenesis in the animal model based on the development of dilated CD31-negative blood vessels in the treated tumors, which is consistent with previous observations of SHetA2 anti-angiogenesis activity in kidney cancer xenografts [15]. CD31, also known as platelet endothelial cell adhesion molecule-1 (PECAM-1), is highly expressed on endothelial cell intercellular junctions where it plays a vital role in regulation of the vascular barrier [30]. CD31 has both cellular adhesion and intracellular signaling properties, which if lost can interfere with junctional integrity and consequently angiogenesis [15]. The localization of dilated CD31-negative vessels adjacent to and within areas of necrosis and apoptosis indicate that SHetA2 and palbociclib inhibition of angiogenesis and the resulting necrosis and apoptosis is contributing to the decreased tumor growth. Although CDK6 has been shown to promote angiogenesis [31], ours is the first study to document in vivo anti-angiogenesis activity of palbociclib. While the mechanism of SHetA2 anti-angiogenesis activity has been shown to occur through reduced production of angiogenic cytokines by cancer cells and direct cell cycle arrest in endothelial cells [15], the mechanism of palbociclib anti-angiogenic activity has yet to be explored. We predict that palbociclib inhibition of CDK4/6 activity in endothelial cells contributes to the anti-angiogenic mechanism.

Consistently with previous studies [17,18,19,32,33], SHetA2 did not cause toxicity in the animal models except for elevation of AST in two mice. AST elevation was not observed in previous studies of SHetA2 [17], and there were no effects of any drug treatments on animal body weights, or on organ-to-body weight ratios, which reduces concern regarding potential toxicity. The most common side effect requiring dose-reduction of palbociclib is neutropenia [20,34]. Our calculated DRIs suggest that, when combined with SHetA2, lower doses of palbociclib could be used to achieve the same level of anti-cancer activity with reduced risk of developing palbociclib-induced neutropenia.

## 4. Materials and Methods

### 4.1. Cell Cultures

The human cervical cancer cell lines SiHa, CaSki, and C33A were obtained from American Type Culture Collection (ATCC, Manassas, VA, USA) and authenticated using autosomal short tandem repeats (STR) profiling by the University of Arizona Genomics Core. All cell lines were maintained with RPMI 1640 with L-glutamine and sodium bicarbonate, liquid, sterile-filtered (R8758, Sigma-Aldrich Saint Louis, MO, USA), with 10% fetal bovine serum (Serum Source International- FBS17712), and 1% antibiotic-antimycotic solution (ABL02- 100X, Caisson Labs, Smithfield, UT, USA). 

### 4.2. Drugs 

Palbociclib HCl (PD-0332991, Cayman Chemical Company, Ann Arbor, MI, USA) and SHetA2 (synthesized by K. Darrell Berlin [Oklahoma State University] according to published methods [35]) were dissolved in dimethylsulfoxide (DMSO, Neta Scientific, Hainesport, NJ, USA) for a stock concentration of 10 mM that was used in tissue culture studies. Palbociclib (P-7788, hydrochloride salt >99%, LC Laboratories, Woburn, MA, USA) and SHetA2 (NSC 726189, provided by the US National Cancer Institute RAPID Program) were suspended in 30% Kolliphor HS 15 (SigmaAldrich, Merck KGaA, Darmstadt, Germany) in water for use in the animal model.

### 4.3. MTT- Cytotoxicity Assay and Drug Interaction Analysis

Cells cultured for 24 h in 96-well plates were treated in a two-fold dilution series, in triplicate with either SHetA2, palbociclib, or a 1:1 combination of their half maximal inhibitory concentrations (IC_50′_s) for 72 h, after which a CellTiter 96^®^ Non-Radioactive Cell Proliferation Assay (MTT) (Promega, Madison, WI, USA) was performed. The fold effect of each drug treatment was calculated by dividing the average corrected OD of the treated cultures from the average corrected OD of control cultures treated with solvent only and then subtracting this quotient from the number 1. Fold effects of the drugs were entered into CompuSyn Software (CompuSyn, New York, NY, USA) for production of isobolograms and derivation of CIs and DRIs. Each analysis was repeated at least 3 times and only curves with r values greater than 0.9 were included in the analysis.

### 4.4. Western Blots

Whole cell protein extracts were prepared using M-PER mammalian protein extraction reagent (ThermoFisher, Grand Island, NY, USA) and volumes corresponding to 20–45 μg determined with Bicinchoninic Acid (BCA) reagent (ThermoFisher, Grand Island, NY, USA) were electrophoresed into a 12% SDS-PAGE gel and transferred to polyvinylidene fluoride or polyvinylidene difluoride (PVDF) membranes. The membranes were blocked for 1 h to overnight with Tris Buffered Saline with Tween (TBST) containing 5% dry skim milk, then washed 3× with TBST and subsequently probed overnight with the following primary antibodies and dilutions: cyclin D1 (#2922, 1:500), phospho-Rb [Ser807/811] D20B12 XP^®^ Rabbit mAb (#8516, 1:1000), alpha tubulin (#2125, 1:1000) all purchased from Cell Signaling Technology (Danvers, MA, USA). Membranes were then washed and incubated with anti-mouse horse radish peroxidase (HRP) conjugated (Santa Cruz Biotechnology, Dallas, TX, USA) or anti-Rabbit HRP conjugated (Cell Signaling Technology, Danvers, MA, USA) for 1 h followed by additional washing and development of signal with enhanced chemiluminescence (ECL) reagents (BioRad, Hercules, CA, USA). Bands were imaged and quantified (ChemiDoc imaging system with Image Lab Software/BioRad).

### 4.5. ELISA

Whole cell protein extracts were prepared using M-PER mammalian protein extraction reagent (ThermoFisher, Grand Island, NY, USA) and volumes corresponding to 0.1–0.4 μg/100 μL determined with BCA reagent (ThermoFisher Grand Island, NY, USA) were evaluated in duplicate or triplicate using the cyclin D1 PathScan^®^ Sandwich ELISA (Cell Signaling Technology, Danvers, MA, USA) and a Synergy H1 microplate BioTek reader (BioTek, Winooski, VT, USA). All analyses were evaluated using protein extracts from three independent experiments.

### 4.6. Animal Model

All of the animal investigations followed the guidelines that were required for the care and use of laboratory animals and were approved by the University of Oklahoma Health Sciences Center institutional animal care and use committee (IACUC Protocol #19-009-CHI). Both SHetA2 and palbociclib were suspended in 30% Kolliphor HS15 (SigmaAldrich, Merck KGaA, Darmstadt, Germany) in water. Four-week old female athymic Hsd:Athymic Nude-*Foxn1^nu^* mice (ENVIGO, Alice, TX, USA) were allowed to acclimate for 2 weeks and then subcutaneously injected with 1 × 10^7^ SiHa cells suspended in 1× phosphate buffered saline (PBS). Tumor sizes were measured with calipers and treatment initiated when tumors achieved ~100mm^3^ average tumor volume ([width^2^ × length]/2). Mice were randomized into 9 animals per treatment group based on tumor volume so that there were no significant differences between the groups (ANOVA, *p* > 0.05). The untreated control group was gavaged with placebo (30% Kolliphor HS 15 in water). The other treatment groups were 60 mg/kg SHetA2, 100 mg/kg palbociclib or the drug combination, and all drug treatments were administered by oral gavage every day for 24 days. Tumors, kidney, livers and spleens were collected and weighed at necropsy. Plasma prepared from the blood collected at necropsy was evaluated liver toxicity markers (ALKP and AST) and kidney function markers (CREA and BUN) using the IDEXX Catalyst instrument following manufacturer’s instructions as described previously [36].

### 4.7. Histochemical and Immunohistochemical Analysis of Xenograft Tumors 

Tumors were fixed in formalin, paraffin-embedded, and 5 micron sections were cut and mounted on positively charged slides. Sections from all tumors were stained with H&E or Trichrome stain. Four tumors with sizes representing each quartile of final tumor volume were selected from each treatment group for immunohistochemical staining using the following primary antibodies and dilutions: cyclin D1 (92G2) (1:50 #2978S, Cell Signaling Technologies, Danvers, MA, USA), phosphorylated Retinoblastoma (1:800, #9516, Cell Signaling Technologies, Danvers, MA, USA), Retinoblastoma (1:200, #B7208, Assay Biotech Fermont, CA, USA), cleaved caspase 3 (1:200, # 9579, Cell Signaling Technologies, Danvers, MA, USA), and CD31 (1:50, # ab28364 Abcam, Cambridge, MA, USA). The immunohistochemistry was performed according to manufacturer’s protocol using Leica Bond-III^TM^ Polymer Refine Detection system (DS 9800). In brief, slides were dried overnight at room temperature and incubated at 60 °C for 45 min followed by deparaffinization and rehydration in an automated Multistainer (Leica ST5020, Leica, Buffalo Grove, IL, USA). Subsequently, these slides were transferred to the Leica Bond-III^TM^, treated for target retrieval at 100 °C for 20 min in a retrieval solution, at a pH of 6.0 for CD31 and cyclin D1 antibodies and a pH 9.0 for cleaved caspase 3, phosphorylated Rb, and total Rb. The sections were incubated with 5% goat serum (01-6201, ThermoFisher Scientific, Grand Island, NY, USA) for 30 min. Endogenous peroxidase was blocked using peroxidase-blocking reagent, followed by incubation with the selected primary antibody for 60 min. For the secondary antibody, post-primary IgG-linker and/or Poly-HRP IgG reagents were used. Detection was done using 3, 3′-diaminobenzidine tetrahydrochloride (DAB), as chromogen and counter stained with hematoxylin. Completed slides were dehydrated (Leica ST5020) and mounted (Leica MM24). Antibody specific positive control (non-experimental tissue chosen for each antigen (cleaved caspase 3: HT-29 xenograft, human malignant stromal tumor, phospho-Rb: human colon carcinoma, Rb: human breast carcinoma and human uterine cancer tissue) and negative control (omission of primary antibody) were stained in parallel. All stained sections were reviewed in a blinded manner by an experienced pathologist (S.L.) who provided pathology evaluation and immunohistochemical stain scores (% cells stained multiplied by intensity of staining).

### 4.8. Statistical Analysis

All experiments were powered to determine if the combined drug treatment was significantly different from the single drug treatments. Comparison of each treatment group with the control group was performed as a secondary analysis. For the cell culture studies, differences between groups was analyzed using ANOVA with Dunnett’s multiple comparison test for normally distributed data in western blot and ELISA results of all three cell lines, with the exception of using the non-parametric Friedman test for the SiHa phospho-Rb western blot results. For the animal model, the repeated measures ANOVA was used with Dunnett’s multiple comparisons test for tumor volume and body weight measurements between the control and treatment groups. To determine if the drugs induced in vivo additive effects, a GEE model was used to analyze tumor growth, which was treated as a continuous variable. Model terms included categorical time, palbociclib (yes/no), SHetA2 (yes/no), interactions between palbociclib and SHetA2, between palbociclib and time, and between SHetA2 and time. The model assumed a compound symmetry covariance structure due to limited sample size. A backward model selection strategy was used such that interaction terms with *p*-values more than 10% were dropped from the model. A linear regression analysis was performed to determine the 95% confidence intervals of the slopes of the lines to compare mouse body growth rates between groups during the treatment periods. The non-parametric Kruskal-Wallis test with Dunnett’s multiple comparisons test was used to compare final tumor weights, organ-to-body weight ratios and immunohistochemical stain scores. The Chi-Square test for trend was used to compare the contingency between the treatments and the phosphor-Rb immunohistochemistry scores. The *p* values for primary analysis and adjusted *p* values for multiple comparisons of <0.05 were considered statistically significant. All analyses, except for the GEE model, were conducted using Graph Pad Prism 8.0 software.

## 5. Conclusions

The results of this “proof of concept” study support targeting the cyclin D1/CDK4/6 complex for development of new treatments for cervical cancer. Combined treatment with drugs that target the two components of this complex, SHetA2 degradation of cyclin D1 and palbociclib inhibition of CDK4/6, exerted synergy in cell culture studies and additive activities in vivo with only inconsistent and minor elevation of toxicity parameters. Complementary inhibition of phospho-Rb in cell lines and SiHa xenograft tumors confirm that the synergy mechanism converges on cyclin D1/CDK4/6 phosphorylation of Rb.

## 6. Patents

Benbrook, D. M., Ramraj, S. “Combination Cancer Therapies,” 62/730,345, United States. (Submitted: 2018).

## Figures and Tables

**Figure 1 cancers-12-01269-f001:**
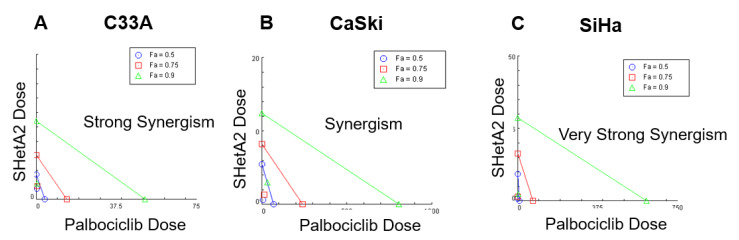
Isobolograms of SHetA2 and palbociclib treatment on (**A**) C33A, (**B**) CaSki, and (**C**) SiHa cervical cancer cell lines.

**Figure 2 cancers-12-01269-f002:**
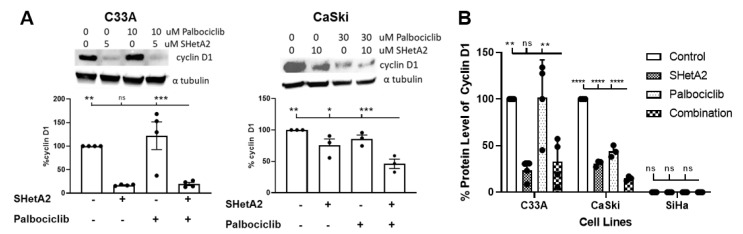
Cyclin D1 levels in cultures treated with the indicated concentrations of SHetA2, palbociclib or the combination of these two drug for 24 h. (**A**) Representative western blots of C33A and CaSki cervical cancer cell lines are shown on top and graphs of cyclin D1 band intensities in replicate experiments are shown below. (**B**) Average of three independent cyclin D1 ELISA measurements in the indicated cell lines treated with the indicated drugs. ANOVA comparison of combination treatment with other treatment groups: ns = not significant, *p* value >0.05 and <0.01, * = *p* < 0.05, ** = *p* < 0.01, *** *p* < 0.001, **** *p* < 0.0001. The whole Western Blot see Appendix A.

**Figure 3 cancers-12-01269-f003:**
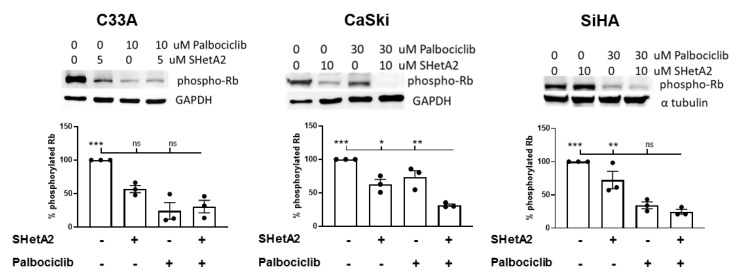
Phospho-Rb levels in cultures treated with the indicated concentrations of SHetA2, palbociclib or the combination of these two drug for 24 h. Representative western blots of the indicated cervical cancer cell lines are shown on top. Graphs of phospho-Rb band intensities in replicate experiments are show below. ANOVA comparison of combination treatment with other treatment groups: ns = not significant, * = *p* < 0.05, ** = *p* < 0.01, *** *p* <0.001. The whole Western Blot see Appendix A.

**Figure 4 cancers-12-01269-f004:**
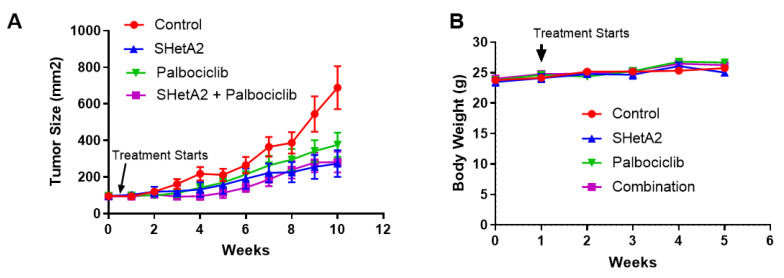
Effect of control, SHetA2, palbociclib, and combination treatments on average tumor size (**A**) and body weight (**B**) in SiHa xenograft tumor bearing mice.

**Figure 5 cancers-12-01269-f005:**
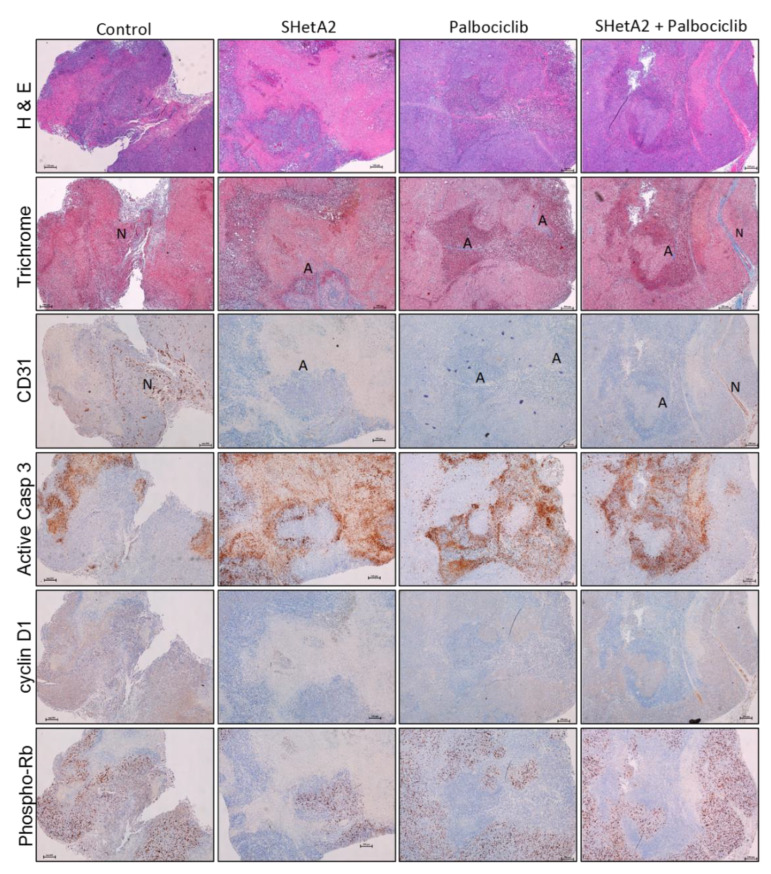
Effects of SHetA2 and palbociclib on SiHa xenograft tumor histology and target proteins. Representative sections of from the indicated treatment groups stained with H&E (Top row), trichrome stain (second row), CD31 (third row), cleaved caspase 3 (Active Casp 3), fourth row), cyclin D1 (fifth row), and phospho-Rb, sixth row). A = Abnormal dilated blood vessels, N = Normal blood vessel. The black line in each of the figures indicates 100 μm.

**Figure 6 cancers-12-01269-f006:**
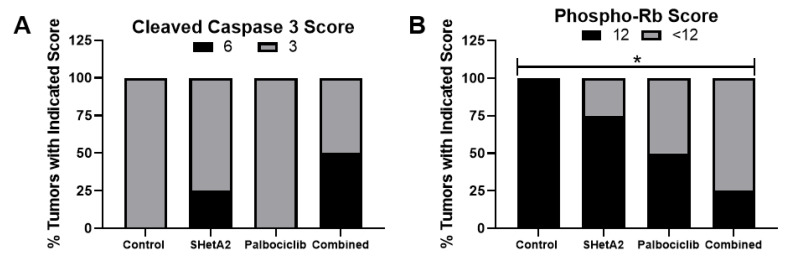
Stain scores of (**A**) cleaved caspase 3 and (**B**) phosphorylated pRb expression in SiHa xenograft tumors after the indicated treatments. Chi-Square Test for Trend * *p* = 0.03.

**Table 1 cancers-12-01269-t001:** Chou-Talalay method results of SHetA2 and palbociclib combination.

Cell Line	IC_50_	DRI		CI		Interpretation
		IC_50_	IC_95_	IC_50_	IC_95_	
SiHa	5.0 µM SHetA2 11.5 µM palbociclib	11.03 µM 8.73 µM	11.47 µM 580.40 µM	0.205	0.089	Very Strong Synergism
CaSki	5.0 µM SHetA2 13.0 µM palbociclib	8.75 µM 9.60 µM	3.07 µM 28.57 µM	0.218	0.361	Synergism
C33A	1.6 µM SHetA2 5.2 µM palbociclib	2.29 µM 6.27 µM	8.31 µM 145.90 µM	0.205	0.116	Strong Synergism

**Table 2 cancers-12-01269-t002:** Kidney and Liver Function Tests in the SiHa Xenograft Mouse Model.

End Point	Normal	Control	60 mg/kg SHetA2	100 mg/kg Palbociclib	SHetA2 + Palbociclib	ANOVA
BUN (mg/dL)	18–29	15.67 ± 1.53	19.33 ± 3.215	12 ± 6.055	10.50 ± 1.291	F = 3.767, *p* = 0.048
CREA(mg/dL)	0.2–0.8	0.045 ± 0	0.197 ± 0.263	0.273 ± 0.263	0.12 ± 0.05	F = 0.0580, *p* = 0.984
ALKP(U/L)	62–209	14.8 ± 17.89	25.17 ± 17.89	27.75 ± 15.5	20 ± 17.89	F = 0.0784, *p* = 0.970
AST(U/L)	59–247	131 ± 59.65	**264 ± 138.7**	74.75 ± 21.82	41.75 ± 18.77	F = 6.499, *p* = 0.0074

Levels above normal are bolded and levels below normal are italicized.

**Table 3 cancers-12-01269-t003:** Organ to Body Weight Ratios of Mice in the Different Treatment Groups.

Organs	Control	60 mg/kg SHetA2	100 mg/kg Palbociclib	Combination	Kruskal-Wallis Test
Kidney	0.017 ± 0.001	0.018 ± 0.002	0.016 ± 0.002	0.010 ± 0.002	*p* = 0.850
Liver	0.053 ± 0.002	0.056 ± 0.007	0.050 ± 0.002	0.050 ± 0.008	*p* = 0.246
Spleen	0.012 ± 0.001	0.0125 ± 0.001	0.010 ± 0	0.010 ± 0	*p* = 0.281

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
