# Peer review of "Complementary Targeting of Rb Phosphorylation and Growth in Cervical Cancer Cell Cultures and a Xenograft Mouse Model by SHetA2 and Palbociclib"

_cancers, 2020, doi:10.3390/cancers12051269_

Round 1

Reviewer 1 Report

The manuscript of Kennedy et al. describes an analysis of two cell cycle inhibitors on cervical cancer cell lines. The authors document that these inhibitors have in vitro activity, and not highly toxic to mice and also have in vivo activity. The paper is well written and referenced and the results support the conclusions.

The paper could be improved as follows:

Major points

The major rationale for this work is to address the effects of the E6 and E7 oncogenes on cell cycle regulators. The authors should address why the compounds work equally well in C33-A cells that do not have HPV sequences. I did find in the COSMIC database that C33-A has a homozygous RB1 gene mutation

Minor points

It should be Chi-square, not Chai.

L 65 and elsewhere. When referring to the gene it should be italicized.

Palbociclib is capitalized sometimes and not others.

You use both Caski and CaSki.

Figure 6 legend, Please use a proper number of significant figures. P=0.03 would be appropriate.

Author Response

We thank Reviewer 1 for careful review of this manuscript and have addressed the concerns as described below.

Major points

The major rationale for this work is to address the effects of the E6 and E7 oncogenes on cell cycle regulators. The authors should address why the compounds work equally well in C33-A cells that do not have HPV sequences. I did find in the COSMIC database that C33-A has a homozygous RB1 gene mutation.

To address the HPV status, we added the following sentence to the discussion on line 223:

We did not evaluate sufficient numbers of HPV+ versus HPV- cell lines to make conclusions about the role of HPV in the drug synergy mechanism.  However, the similar responses of the HPV- C33A cell line to the HPV+ cell lines suggests that the evolution of the originating tumors for these cell lines resulted in similar defects that make them susceptible to SHetA2 and palbociclib. Thus, the SHetA2 and palbociclib combination may have efficacy in non-HPV cancers.

The homozygous Rb mutation listed in COSMIC for the RB1 gene is described as not translating into an amino acid mutation, so it will not affect the protein function.

Minor points

It should be Chi-square, not Chai.

L 65 and elsewhere. When referring to the gene it should be italicized.

Palbociclib is capitalized sometimes and not others.

You use both Caski and CaSki.

Figure 6 legend, Please use a proper number of significant figures. P=0.03 would be appropriate.

Thank you for pointing these out. We have corrected all of these points in revisions of the text within the figures, figure legends and manuscript text where appropriate. We have also updated the figure font and size to be consistent with the rest of the manuscript.

Reviewer 2 Report

The authors here make an effort to use the combination of a CDK4/6 inhibitor pablociclib and Cyclin D1 inhibitor SHetA2 to treat cervical cancer. However, there are a few major concerns.

  1. The authors state that SiHa cells do not express Cycli D1 enough to be detected by either western blot or ELISA. How do they then explain the significant cytotoxicity induced by SHetA2 and strong synergism displayed by the combination?
  2. It might be more convincing to at least confirm the results at RNA levels (Cyclin D1) by PCR for SiHa cells. In figure 5, cyclin D1 has been shown by IHC in SiHa xenografts. If cyclin D1 is detectable by IHC in tumor sections(though it is week), it must be detectable in vitro as well. Please try IFC or ICC if western blot and ELISA do not work.
  3. In vitro, the combination in SiHa cells showed superior response only in comparison to SHetA2 and not pablociclib. Why do the authors then do the in vivo study with SiHa xenografts and how do they explain these results? As such, SiHa cells are known to be weekly tumorigenic. HPV +ve cell lines HeLa and ME180 would have been better choices.
  4. Figure 5, the IHC panel shows trichrome staining but nothing has been discussed about its significance. Stroma plays an important role in tumor progression.
  5. How do the authors identify blood vessels negative for CD31 staining? Do they use any other marker?
  6. Higher magnification IHC images are required.

Author Response

We thank reviewer 2 for providing a detailed critique of our manuscript and believe that by addressing the criticisms, this manuscript has been significantly improved.

  1. The authors state that SiHa cells do not express Cycli D1 enough to be detected by either western blot or ELISA. How do they then explain the significant cytotoxicity induced by SHetA2 and strong synergism displayed by the combination?

We are able to detect a minimal level of cyclin D1 in the immunohistochemical stain of SiHa tumors that is reduced upon SHetA2 treatment.  To explain the toxicity, we have added the following sentences to line 219 of the discussion:

It is likely that additional mechanisms contribute the cytotoxicities of SHetA2 and palbociclib when administered alone and in combination. SHetA2 causes upregulation of intrinsic and extrinsic apoptosis pathway components, degradation of Bcl-2 and translocation of wild type and mutant p53 to the nucleus and mitochondria [14,18,23-27]. Palbociclib has been shown to bind multiple off-target kinases and affect down-stream signaling in vitro and in vivo [28,29].

  1. It might be more convincing to at least confirm the results at RNA levels (Cyclin D1) by PCR for SiHa cells. In figure 5, cyclin D1 has been shown by IHC in SiHa xenografts. If cyclin D1 is detectable by IHC in tumor sections(though it is week), it must be detectable in vitro as well. Please try IFC or ICC if western blot and ELISA do not work.

We have added additional supplementary data and replaced the sentence: “Levels of cyclin D1 in the SiHa cell line were below the level of detection by western blot, and also by the more-sensitive ELISA method.” with the sentence on line 124 and shown below:

The level of cyclin D1 in SiHa was too low to allow accurate measurement of SHetA2 reduction in western blot or ELISA analysis, however western blot analysis confirmed that a low level is present in SiHa cells and can be induced to higher levels by nicotine-derived nitrosamine ketone (NNK) (Supplemental Figure S2).

  1. In vitro, the combination in SiHa cells showed superior response only in comparison to SHetA2 and not pablociclib. Why do the authors then do the in vivo study with SiHa xenografts and how do they explain these results? As such, SiHa cells are known to be weekly tumorigenic. HPV +ve cell lines HeLa and ME180 would have been better choices.

We chose to utilize the SiHa cell line for the xenograft studies, because our review of the literature gave us the impression that this cell line is the most commonly used and accepted model.  In contrast, the HeLa cell line is often criticized for having been passaged so extensively that it may no longer be representative of cervical cancer. We currently do not have the ME180 cell line in our collection, but we appreciated the recommendation and will include this cell line in future studies.

  1. Figure 5, the IHC panel shows trichrome staining but nothing has been discussed about its significance. Stroma plays an important role in tumor progression.

We agree that the trichrome stain can reveal important information about the tumor stroma, however review of the trichrome stained sections by an experienced pathologist did not reveal notable stromal effects to report.  To acknowledge this, we added the following sentences to line 183 of the results section:

No obvious changes in the collagen or cell densities were observed in the stroma of trichrome stained sections.

  1. How do the authors identify blood vessels negative for CD31 staining? Do they use any other marker?

Our primary identification of blood vessels was done using the blue color of the trichrome stain, with CD31 as the additional marker.

  1. Higher magnification IHC images are required.

We reduced the font size in the figure legend to increase the area of the page availale for the figure itself. We chose the magnification used in Figure 5 to show a sufficient portion of the tumor that documents the heterogeneity and consistency of stains within live versus necrotic areas of the tumor.  We believe that this magnification avoids bias of selecting smaller areas that would not reveal the tumor heterogeneity, and believe that this increases the robustness of the data. In addition, this magnification allows visualization of both tumor and adjacent stroma.  Also, we believe that this level of magnification allows sufficient visualization of tissue architecture when the figure is printed or visualized as a full page image. 

Round 2

Reviewer 2 Report

The authors have tried to address most of the concerns, however, they have only modified the write up and have not included any new experimental data. The paper is technically sound, but in my opinion, does not seem to be insightful enough to be published in Cancers.